# Conformational Models of APP Processing by Gamma Secretase Based on Analysis of Pathogenic Mutations

**DOI:** 10.3390/ijms222413600

**Published:** 2021-12-18

**Authors:** Meewhi Kim, Ilya Bezprozvanny

**Affiliations:** 1Department of Physiology, UT Southwestern Medical Center at Dallas, Dallas, TX 75390, USA; 2Laboratory of Molecular Neurodegeneration, Peter the Great St. Petersburg State Polytechnical University, 195251 St. Petersburg, Russia; 3Laboratory of Synaptic Biology, Southern Federal University, 344006 Rostov-on-Don, Russia

**Keywords:** gamma-secretase, APP, Alzheimer’s disease, modeling, protein disorder

## Abstract

Proteolytic processing of amyloid precursor protein (APP) plays a critical role in the pathogenesis of Alzheimer’s disease (AD). Sequential cleavage of APP by β and γ secretases leads to the generation of Aβ40 (non-amyloidogenic) and Aβ42 (amyloidogenic) peptides. Presenilin-1 (PS1) or presenilin-2 (PS2) play the role of a catalytic subunit of γ-secretase. Multiple familial AD (FAD) mutations in APP, PS1, or PS2 result in an increased Aβ42:Aβ40 ratio and the accumulation of toxic Aβ42 oligomers and plaques in patient brains. In this study, we perform molecular modeling of the APP complex with γ-secretase and analyze potential effects of FAD mutations in APP and PS1. We noticed that all FAD mutations in the APP transmembrane domain are predicted to cause an increase in the local disorder of its secondary structure. Based on structural analysis of known γ-secretase structures, we propose that APP can form a complex with γ-secretase in 2 potential conformations—M1 and M2. In conformation, the M1 transmembrane domain of APP forms a contact with the perimembrane domain that follows transmembrane domain 6 (TM6) in the PS1 structure. In conformation, the M2 transmembrane domain of APP forms a contact with transmembrane domain 7 (TM7) in the PS1 structure. By analyzing the effects of PS1-FAD mutations on the local protein disorder index, we discovered that these mutations increase the conformational flexibility of M2 and reduce the conformational flexibility of M1. Based on these results, we propose that M2 conformation, but not M1 conformation, of the γ secretase complex with APP leads to the amyloidogenic (Aβ42-generating) processing of APP. Our model predicts that APP processing in M1 conformation is favored by curved membranes, such as the membranes of early endosomes. In contrast, APP processing in M2 conformation is likely to be favored by relatively flat membranes, such as membranes of late endosomes and plasma membranes. These predictions are consistent with published biochemical analyses of APP processing at different subcellular locations. Our results also suggest that specific inhibitors of Aβ42 production could be potentially developed by selectively targeting the M2 conformation of the γ secretase complex with APP.

## 1. Introduction

Alzheimer’s disease (AD) is a major health problem for developed nations that has, so far, resisted the development of effective therapies. Then amyloid hypothesis of AD indicates that the accumulation of amyloidogenic Aβ42 peptides is a major driving force responsible for AD [1,2,3,4]. Both the amyloidogenic Aβ42 peptide and the non-amyloidogenic Aβ40 peptide are generated as a result of sequential proteolytic cleavage of amyloid precursor protein (APP) by β and γ secretases [5,6]. The amyloid hypothesis postulates that an increased ratio of Aβ42:Aβ40 levels is a key pathogenic event in AD [1,2,3,4]. γ-Secretase is a multiprotein membrane complex composed of nicastrin, presenilin enhancer 2 (Pen-2), anterior pharynx defective 1 (Aph1), and presenilin [7,8,9]. In addition to APP, γ-secretases cleave various type I transmembrane proteins, including the Notch receptor [10]. Most likely because of these additional substrates, pharmacological targeting of γ-secretase so far has failed to yield effective therapies for AD [11,12]. Multiple clinical trials of γ-secretase inhibitors have failed partly due to side effects resulting from the inhibition of cleavage of the Notch receptor and other γ-secretase substrates, such as, for example, the trial of semagacestat (LY-450139) [13]. Thus, there is a significant effort by the industry to develop “Notch-sparing” γ-secretase inhibitors that selectively block the generation of Aβ42 but do not affect Notch receptor cleavage. Examples of such Notch-sparing compounds are avagacestat (BMS-708163) [14] and begacestat (GSI-953) [15]. However, avagacestat also failed in AD clinical studies [16,17], and begacestat has not moved past a phase 1 trial in clinical development [18]. Moreover, the Notch-sparing selectivity of avagacestat was questioned [19]. 

To inform this important direction of research, we here perform molecular modeling of the APP complex with γ-secretase. In our modeling, we rely on γ-secretase structures that have been recently solved by the application of cryo-EM [20,21]. Familial AD (FAD) mutations in APP or presenilins are known to have significant effects on the production of Aβ42 and Aβ40 peptides and the Aβ42:Aβ40 ratio [22,23,24,25]. We analyzed the potential effects of these mutations on the conformational flexibility of the APP complex with γ-secretase. Based on this analysis, we propose the existence of two alternative conformations of the APP complex with γ-secretase—M1 and M2. Furthermore, our analysis suggests that the APP cleavage in M1 conformation favors Aβ40 production, whereas APP in M2 conformation favors Aβ42 production. Structural modeling also suggested that APP processing in M1 conformation is favored by curved membranes, such as the membranes of early endosomes. In contrast, APP processing in M2 conformation is likely to be favored by relatively flat membranes, such as membranes of late endosomes and plasma membranes. These predictions are consistent with the biochemical analysis of APP processing in different subcellular locations [26,27,28,29,30,31]. The proposed co-existence of M1 and M2 complexes of APP and γ-secretase may help structure the functional analysis of APP processing and may potentially facilitate the development of more selective Notch-sparing inhibitors of γ-secretase for therapeutic use. 

## 2. Results

### 2.1. AD Mutations Destabilize the Secondary Structure of the APP Transmembrane Domain

APP is a 770 amino-acid protein with a single transmembrane domain in the carboxy-terminal end (Figure 1A). In the process of generating Aβ peptides, APP is cleaved initially by a β-secretase at position 671 and then by a γ-secretase within a transmembrane domain (Figure 1A). There is a number of AD-causing mutations in the APP sequence, most of them located within the APP transmembrane domain (Figure 1A) [22,23,24,25]. In order to understand the potential impact of these mutations on APP secondary structure, we calculated changes in the local disorder index (DI) resulting from these mutations using the DISOPRED3 web-based program [32] (see Methods for details). For this calculation, we utilized a fragment of APP sequence (671–730) that starts at the β-secretase cleavage site and ends several amino acids after the transmembrane domain (Figure 1A). In our calculations, this fragment was modeled as a continuous α-helix. The AD-causing mutations in the transmembrane domain of APP are located between amino acids 705 and 723 (Figure 1A). In some of these positions, only a single substitution (MUT1) is known to be pathogenic, but in some positions, several pathogenic substitutions have been reported. In these positions, we performed calculations for one more potential substitution (MUT2). The AD-causing mutations that we analyzed were as follows—L705V, A713V, T714A (Iranian), T714I (Austrian), V715M (French), V715A (German), I716V (Florida), I716F (Iberian), V717F (Indiana), V717I (London), and T719N, T719P, M722K, and L723P (Australian) [22,23,24,25]. This analysis revealed that all AD-causing mutations in the transmembrane domain of APP significantly destabilize the secondary structure of the APP domain, with the changes in the local disorder index in the range of 10–12 for most mutations (Figure 1B). Based on these results, we concluded that the destabilization and partial unwinding of the α-helical structure of the APP transmembrane domain can facilitate the amyloidogenic processing of APP by γ-secretase and the generation of the Aβ42 peptide. 

### 2.2. Protein Structural Models of the APP Complex with γ-Secretase

To better understand the mechanistic basis of these results, we generated a structural model of the APP complex with γ-secretase using γ-secretase structures that have been recently solved by the application of cryo-EM [20,21]. Presenilin-1 (PS1) or presenilin-2 (PS2) are catalytic subunits of γ-secretase [7,8,9]. In our modeling, we focused on PS1. The matured PS1 structures within γ-secretase are heterodimers of PS1-NT (TM1–TM6) and PS1-CT (TM7–TM9) fragments. We downloaded all γ-secretase structures from the Protein Data Bank (PDB_IDs 6IDF, 6IYC, 5A63, 5FN2,5FN3, 5FN4, 5FN5, 4UIS) and closely examined the conformation of the γ-secretase active site formed by TM6 and TM7 transmembrane domains of PS1. The reported PS1 structures miss most of the soluble domains, with the exception of a small portion of a perimembrane region that follows TM6. The structure of this domain is solved in different conformations, ranging from unstructured conformation to a loop and α-helix near the membrane [20,21]

We found that three TM helixes of PS1-CT (TM7–TM9) are largely overlapping in all structures, but the TM helixes of PS1-NT (TM1–TM6) adopt different conformations in different structures (Figure 2A compares structures from 5FN4, 5FN5, and 6IYC). In particular, the TM6 helix is found in two different conformations, as shown in Figure 2A. We used these structures to model possible locations of the APP transmembrane domain α helix within an active site of a γ-secretase. As a result of the model building, we were able to produce two possible protein models of the complex—Model 1 (M1, based on PDB 6IYC) (Figure 2B) and Model 2 (M2, based on 5FN3) (Figure 2C). In designing these models, we made sure that the transmembrane domain of APP was in direct contact with the γ-secretase active site formed by the D257 (TM6) and D385 (TM7) residues of PS1. In model M1, APP forms a contact with the perimembrane domain that follows TM6 in the PS1 structure (based on PDB 6IYC) (Figure 2B). In model M2, APP forms a contact with TM7 domains (based on PDB 5FN3) (Figure 2C). Both models were optimized to satisfy overall protein–protein interactions from polar, nonpolar, and aromatic residues (see Methods for details).

### 2.3. Effects of AD Mutations in PS1 on the Stability of the APP Complex with γ-Secretase

PS1 contains 9 TMs (Figure 3A) with AD-causing mutations spread across all of them [23,25]. We focused on PS1 for this analysis as structural information about PS2 is not available. Structural modeling suggests that TM6 and TM7 of PS1 form direct contact with the APP transmembrane domain (Figure 2). What is the influence of the mutations within TM6 and TM7 domains on the association between APP and γ-secretase? To answer this question, we calculated the changes in the local disorder index (DI) resulting from PS1 mutations by using the same approach described above for the APP transmembrane domain (see Methods for details). For this analysis, we selected a region between A251 and A400 of PS1 (Figure 3A, mutations shown in bold font). This region is directly involved in molecular interactions with APP (Figure 2) and includes a part of the TM6 domain, the large soluble region between TM6 and TM7 domains that contains the endoproteolysis cleavage site (at position L286) and the TM7 domain (Figure 3A). D257 and D385 aspartates that form the catalytic site of γ-secretase are both located within this region (Figure 3A). In our analysis, we excluded a large portion of the soluble loop between TM6 and TM7 domains (positions between 287–363) as a secondary structure of this region is not resolved by cryoEM and it is not involved in significant interactions with APP based on our model (Figure 2). To separate the effects of the mutations in TM6 and TM7 domains in our calculations, we divided the relevant PS1 sequence into two regions—D1 and D2 (Figure 3A). The D1 region included mutations in positions Y256-A285, and the D2 region included mutations in positions S365-A396 (Figure 3A). To estimate the change in the local disorder index, additional flanking transmembrane sequences from TM6 and TM7 were also included in the calculations (Figure 3A). The results of these calculations for the representative AD-causing mutations in the region between Y256 and A396 are shown in Figure 3B. Similar results were obtained when other pathogenic mutations of PS1 in the same positions were evaluated (data not shown). It is noticeable that most AD-causing mutations in the D1 region resulted in a negative change in DI, consistent with a reduction in conformational flexibility of the local PS1 structure (Figure 3B). The exceptions to this rule are mutations A260V and V261F, which resulted in a positive change in DI (Figure 3B). In contrast, most AD-causing mutations in the D2 region resulted in a positive change in DI, consistent with increased conformational flexibility of the PS1 structure (Figure 3B). The exceptions to the latter rule are mutations G378V, G384A, and Y389F, which resulted in a negative change in DI (Figure 3B). 

It is interesting to compare these results of the structural modeling of the APP complex with γ-secretase (Figure 2). The PS1 region corresponding to D1 is involved in direct association with the transmembrane domain of APP according to the M1 model of the complex (Figure 2B,C) but not according to the M2 model (Figure 2D,E). In contrast, the PS1 region corresponding to D2 is involved in association with the transmembrane domain of APP according to the M2 model of the complex (Figure 2D,E) but not according to the M1 model (Figure 2B,C). Therefore, it appears that most disease-causing mutations reduce the conformational flexibility of the γ-secretase complex with APP in M1 conformation but increase the conformational flexibility of the γ-secretase complex with APP in M2 conformation. Our previous analysis of AD-causing mutations in the APP transmembrane domain revealed that all of these mutations increased the disorder index and conformational flexibility of APP (Figure 1B). To explain these findings, we propose that PS1 mutations in the D2 region act similar to APP mutations by promoting the conformational flexibility of the complex in M2 conformation, resulting in the increased generation of Aβ42. In contrast, we reason that PS1 mutations in the D1 region act by reducing the conformation flexibility of the complex in M1 configuration, resulting in the impaired production of Aβ40. An overall result of all these mutations (in APP and in the D1 and D2 regions of PS1) is an increased ratio of Aβ42:Aβ40 produced by the γ-secretase. An interesting exception to this rule is the five mutations in the D1 and D2 regions discussed above. It is striking that four out of these five mutations are located within immediate proximity of catalytic aspartates D257 and D385 in the secondary structure of PS1 (Figure 3C,D). Thus, most likely, these four mutations have a direct impact on the catalytic reaction mediated by D257 and D385, in contrast to other mutations in the D1 and D2 regions that, most likely, act by affecting overall γ-secretase movement during APP proteolysis. 

### 2.4. Effects of Membrane Curvature on APP Processing by γ-Secretase

Processing of APP by γ-secretase occurs in plasma membranes and in early and late endosomal compartments [26,28,29,30,31]. Plasma membranes are relatively flat, but endosomal membranes are curved. Because of the difference in size, membranes of early endosomes (typical diameter from 40 to 100 nm) are significantly more curved than membranes of larger (typical diameter from 250 to 400 nm) and later endosomes [33]. What effect, if any, can the curvature of the membrane have on APP processing by γ-secretase? To answer this question, we modeled the APP/γ-secretase complex in M1 and M2 conformations in the membranes with different curvatures. Based on the known cryo-EM APP/γ-secretase complex structure [21], in the first approximation, we modeled PS1 embedded in the flat plasma membrane as a cylinder with a diameter of 0.40 nm and height of 0.45 nm (average size measured from PS structures of PDBs) (Figure 4A). We further reasoned that as a result of membrane expansion, the shape of PS1 embedded into the curved membrane would be affected by an expansion on the cytosolic side by a distance equal to d. From our estimates, this distance for early endosomes (d1) is in the range between 4Å and 12Å, depending on the size of the endosome (Figure 4B). For late endosomes, this distance (d2) is estimated to be less than 1Å (Figure 4C). We further reasoned that these differences in PS1 geometry may have differential effects on APP processing by γ-secretase in M1 and M2 conformations. 

Structural modeling suggests that the perimembrane region D1 that follows the TM6 region of PS1 is involved in direct association with the APP transmembrane domain in M1 conformation of the APP/γ-secretase complex (Figure 2B). Our hypothesis is that APP cleavage by γ-secretase in M1 conformation involves the rotation of TM6 and D1 regions (Figure 4D), and such movement should be facilitated when PS1 is embedded in the curved membrane with an expanded cytosolic interface. In contrast, inner membrane region D2 of PS1-TM7 is involved in direct association with the transmembrane domain of APP in M2 conformation of the γ-secretase complex (Figure 2D). We propose that APP cleavage by γ-secretase in M2 conformation involves the stretching of the TM7 region perpendicular to the membrane surface (Figure 4E) and that such movement is facilitated when PS1 is embedded into flat membranes or into membranes with low curvature. Based on this analysis, we suggest that APP cleavage by γ-secretase in early endosomal compartments is biased towards M1-mediated processing and Aβ40 production, while APP cleavage by γ-secretase in late endosome compartments and in the plasma membrane is biased towards M2-mediated processing and Aβ42 production. This hypothesis is consistent with the biochemical analysis of APP processing in different subcellular locations [26,28,29,30,31]. 

## 3. Discussion

### 3.1. Comparison of M1 and M2 Models with Known Structures of the APP/γ-Secretase Complex

The structure of the APP/γ-secretase complex was previously solved by cryo-EM (PDB 6IYC) [21]. This structure was critical for building model M1 (Figure 2B). In this structure, the transmembrane domain of the APP peptide (L705-K726) forms molecular interactions with TM2, TM3, TM5, TM6, and TM7 of PS1. The secondary structure of the APP protein in structure 6IYC makes a transition from α to β conformation at the carboxy terminal end, where it forms an antiparallel β-strand interaction with the β-strand of TM7 of PS1. Similar to this structure, our models also indicate that the APP substrate accesses the γ-secretase active site by interacting with TM5, TM6, and TM7 of PS1 (Figure 2B,D). However, our models make different predictions regarding the conformational movements of γ-secretase during the catalytic reaction of APP hydrolysis. Cryo-EM structure (PDB 6IYC) suggests that catalytic reaction involves the interaction of the α-helical amino-terminal APP region with TM2 and TM3 of PS1 and the β-strand association of the carboxy-terminal APP region with TM7 [21]. In contrast, we modeled the association of the APP α-helix with D1-TM6 (in M1 conformation; Figure 2B,C) or with D2-TM7 (in M2 conformation; Figure 2D,E). Based on these structural models, we predicted that APP cleavage by γ-secretase in M1 conformation involves the rotation of TM6 and D1 regions (Figure 4D) and that APP cleavage by γ-secretase in M2 conformation involves the stretching of the TM7 region perpendicular to the membrane surface (Figure 4E). 

Our hypothesis is consistent with the “open” and “closed” conformations of PS1 that have been previously proposed based on FRET-based studies of PS1 conformation [34]. It has been suggested that the “open” conformation of PS1 correlates with low Aβ42/40 ratios and the “closed” conformation of PS1 correlates with high Aβ42/40 ratios [34]. Thus, it is plausible that M1 conformation of γ-secretase (Figure 2B,C) corresponds to an “open” conformation of PS1 and the M2 conformation of γ-secretase (Figure 2D,E) corresponds to a “closed” conformation of PS1, as described by these authors. 

Our hypothesis is also consistent with biochemical analysis of APP processing in different subcellular locations. Indeed, it has been reported that late endosome compartments contain Aβ peptides with a higher 42/40 ratio [26,27,28,29,30,31]. Mechanistic studies of γ-secretase activity have indicated increased Aβ42 recycling from early to late endosomes [26]. Noticeably, these studies were primarily focused on the influence of endosomal pH on γ-secretase activity. Our hypothesis regarding the potential effects of membrane curvature has not been previously explored, and it requires further experimental testing. 

It should also be noted that our analysis was primarily focused on the catalytic site of γ-secretase that is composed of TM6 and TM7 domains of PS1. However, many FAD mutations in presenilins are located outside of this region [23,25], and it is possible that mutations in these additional domains also affect the conformational flexibility of the γ-secretase catalytic site. We have not been able to analyze the effects of these mutations as our analysis is limited to the calculations of local protein disorder and is not applicable to an analysis of long-range effects. More sophisticated modeling and molecular dynamics simulation approaches may help address this question by taking advantage of known structures of γ-secretase [20,21].

### 3.2. Potential Implications for the Development of Selective γ-Secretase Inhibitors

The amyloid hypothesis predicts that γ-secretase should be the most promising therapeutic target for the treatment of AD as inhibition of Aβ42 production should stop the progression of AD pathology [1,2,3,4]. However, γ-secretase inhibitors such as, for example, semagacestat (LY-450139), have failed in clinical trials [11,12]. One of the potential reasons for these failures is side effects resulting from the inhibition of cleavage of the Notch receptor and other γ-secretase substrates. To minimize these problems, the industry has been focused on developing “Notch-sparing” γ-secretase inhibitors that selectively block the generation of Aβ42 but do not affect the Notch receptor cleavage. However, such inhibitors, such as, for example, avagacestat (BMS-708163), have also failed in clinical trials [16]. These efforts could be informed by the results of our modeling studies. Our hypothesis suggests that specific inhibitors of M2 conformation of the γ-secretase/APP complex may selectively inhibit the production of Aβ42. Such selective inhibitors may be potentially developed by structure-guided virtual screening and the selection of small molecules that more potently inhibit γ-secretase in M2 conformation (Figure 2D,E) than in M1 conformation (Figure 2B,C). 

## 4. Materials and Methods

### 4.1. Calculations of the Disorder Index of FAD Mutations in APP and PS1

To calculate changes in the local disorder index (DI), we used the web-based program DISOPRED3 [32], available at http://bioinf.cs.ucl.ac.uk/psipred/ (accessed on 4 November 2021) using a fragment of the APP sequence (M671-S730). The AD-causing mutations that we analyzed were as follows—L705V, A713V, T714A (Iranian), T714I (Austrian), V715M (French), V715A (German), I716V (Florida), I716F (Iberian), V717F (Indiana), V717I (London), and T719N, T719P, M722K, and L723P (Australian). Wild-type and single-residue mutated fragments of APP were subjected to DI calculation by DISOPRED3, and the calculated DI for each mutant was subtracted from wild-type DI to obtain ΔDI. 

A similar approach was used in the analysis of PS1 FAD using D1 (A251-L286) and D2 (L364-A400) regions in calculations. To calculate ΔDI, DI was calculated for wild-type D1 and D2 sequences and for a series of FAD mutations in D1 (Y256F, A260V, V261F, L262F, C263F, P264L, G266S, P267T, R269H, L271V, E273A, T274R, A275V, R278T, E280A, L282V, A285V) and D2 (S365Y, L381F, Y389F, R377M, G378V, G384A, F386S, S390I, V391F, L392V, G394V, A396T) to yield ΔDI for each mutation.

### 4.2. Structural Models of the APP Complex with γ-Secretase

The M1 model of the APP complex with γ-secretase is based on PDB 6IYC [21], and the M2 model of the APP complex with γ-secretase is based on PDB 5FN3 [20] The transmembrane domain of APP was modeled as an α-helix. In the first step, we modeled molecular interactions with the APP transmembrane domain with TM5, TM6, and TM7 in PS structures. Because the 6IYC structure was solved in the complex with APP, only a slight adjustment of the structure was required to generate model M1. To generate the M2 model, the PS1-NT structure from 5FN3 was adjusted by rotating further using loop regions without disturbing the active site formed by D257 and D385. Such manipulation is similar to the sliding movement observed for PS1-NT in known structures [PDB_ids 6IDF, 6IYC, 5A63, 5FN2,5FN3, 5FN4, 5FN5, 4UIS, 4HYG, 4HYC, 4HYD]. At the second step, APP in model M1 was modeled to form a contact with the perimembrane domain D1 that follows TM6 in the PS1 structure. APP in model M2 was modeled to form contact with the TM7 domain in the PS1 structure. in the third step for both models, we placed the VIAT sequence of APP at the γ-secretase catalytic site formed by D257 and D385. The APP α-helix was adjusted to have proper interactions with polar, nonpolar, and aromatic residues within TM5, TM6, and TM7 domains to yield the final versions of the M1 and M2 models of the complex. While tuning these interactions, the VIAT residues remained at the active site. 

## Figures and Tables

**Figure 1 ijms-22-13600-f001:**
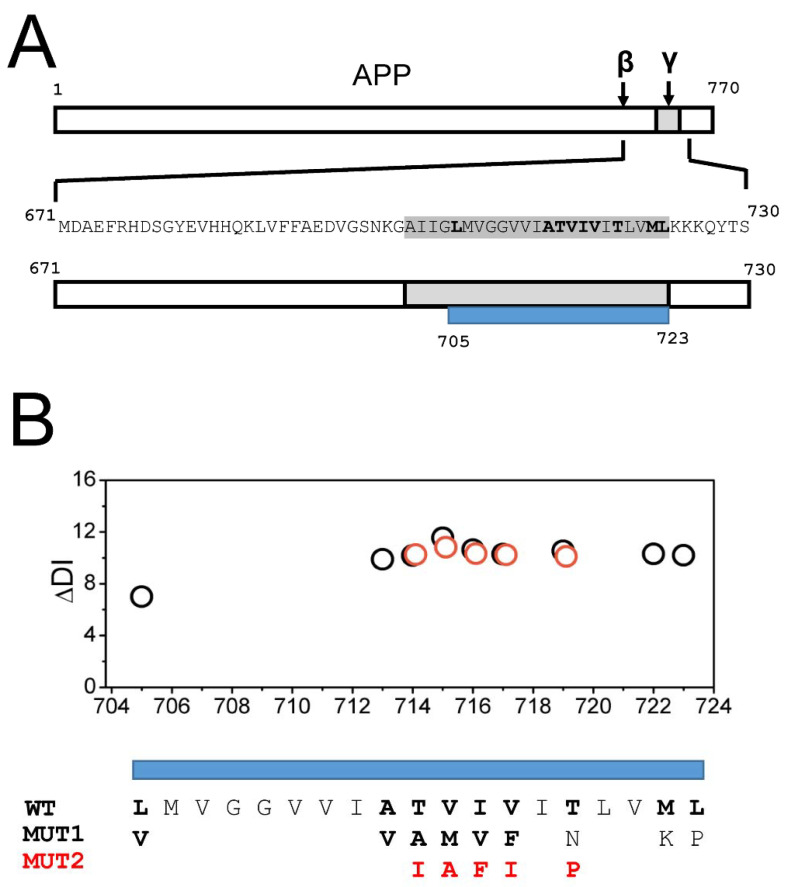
FAD mutations of the APP transmembrane domain destabilize the secondary structure. (**A**) Domain structure of APP, with locations of β- and γ-secretase cleavage sites, is indicated. The sequence of APP from the β-secretase cleavage site (M671) to the soluble region following the transmembrane domain (S730) is indicated. Transmembrane domain is shown in grey; positions that mutated in FAD are shown in bold font. Blue bar shows the boundaries of the mutated region. (**B**) Calculated difference in the disorder index (ΔDI) for FAD mutants is shown for MUT1 (open circles) and MUT2 (red circles).

**Figure 2 ijms-22-13600-f002:**
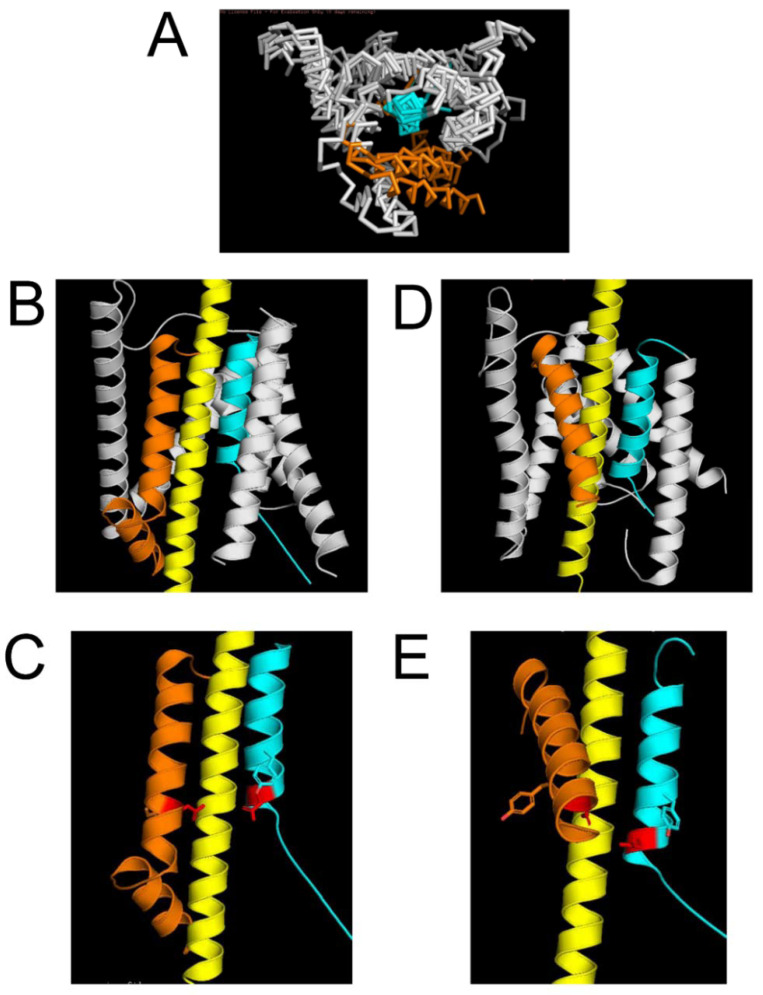
**Structural models of the γ-secretase/APP complex.** (**A**). An overlay of PS1 structures (PDB 5FN4, 5FN5, 6IYC). TM7 is colored in cyanate, and TM6 is colored in orange. All other TMs are colored in grey. (**B**,**C**). M1 model of the γ-secretase complex with APP. D1-TM6 region is colored in orange, TM7 is colored in cyanate, and APP is colored in yellow. Boundaries of the D1 region are shown in Figure 3. Only D1-TM6, TM7, and APP are shown in Panel C for clarity. Catalytic aspartates are colored in red. (**D**,**E**). M2 model of the γ-secretase complex with APP. TM6 region is colored in orange, D2-TM7 is colored in cyanate, and APP is colored in yellow. Boundaries of the D2 region are shown in Figure 3. Only D1-TM6, TM7, and APP are shown on Panel E for clarity. Catalytic aspartates are colored in red.

**Figure 3 ijms-22-13600-f003:**
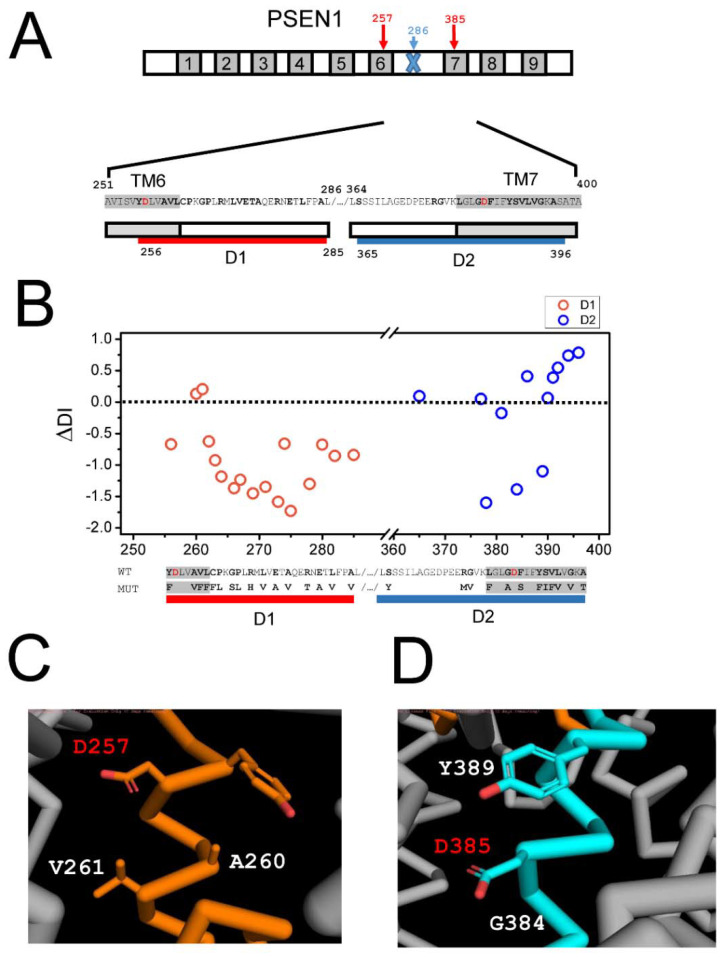
Effects of FAD mutations in PS1 on the stability of the APP complex with γ-secretase. (**A**). Bar diagram shows the domain structure of PS1 with locations of catalytic aspartates (D257 and D385) and the location of endoproteolytic cleavage site (L286) indicated. The sequences of the D1 (A251-L286) and D2 regions (L364-A400) are indicated. Transmembrane domains are shown in grey; positions that mutated in FAD are shown in bold font; catalytic aspartates are shown in red font. Red bar (D1) and blue bar (D2) show the boundaries of the mutated region that was analyzed. (**B**). Calculated difference in the disorder index (ΔDI) for FAD mutants is shown for the D1 region (red circles) and the D2 region (blue circles). (**C**). High-resolution structural model of the γ-secretase active site, with positions of V261 and A260 shown relative to the catalytic aspartate D257. (**D**). High-resolution structural model of the γ-secretase active site, with positions of Y389 and G384 shown relative to the catalytic aspartate D385.

**Figure 4 ijms-22-13600-f004:**
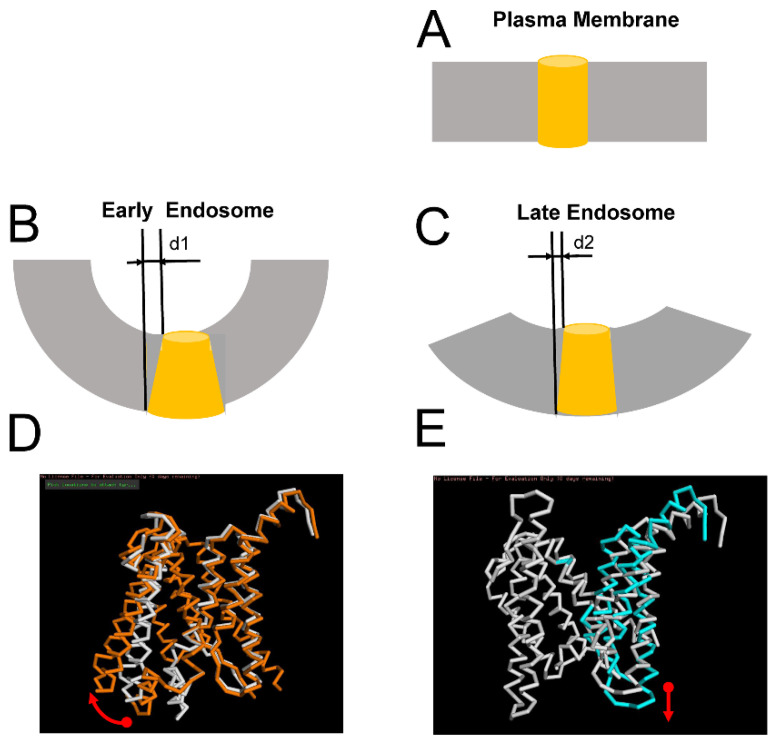
Effects of membrane curvature on γ-secretase activity. (**A**). PS1 is modeled as a cylinder shape in the plasma membrane (**B**,**C**) PS1 modeled as a truncated cone shape in the curved membrane of early (**B**) and late (**C**) endosomes. Expansion of the PS1 shape at the cytosolic surface of the membrane is shown as d1 (**B**) and d2 (**C**). (**D**). APP processing by γ-secretase is modeled as the lateral movement of PS1 in M1 conformation (shown as an overlay). The movement is a 7Å lateral movement between known PS1 structure 6IYC (white) and PS1 conformation in model M1 (orange). (**E**). APP processing by γ-secretase is modeled as a stretching movement of PS1 in M2 conformation (shown as an overlay). The movement is a 3Å expansion of the TM7 domain between the known PS1 structure 5FN3 (white) and PS1 conformation in model M2 (cyanate).

## Data Availability

Not applicable.

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
