# Peer review of "Conformational Models of APP Processing by Gamma Secretase Based on Analysis of Pathogenic Mutations"

_ijms, 2021, doi:10.3390/ijms222413600_

Round 1
Reviewer 1 Report
Aβ plaque and Neurofibrillary tangles are the characteristic neuropathological hallmarks of Alzheimer’s’ disease. Investigators attempt to stop their build-ups in the brain to abate the AD progression or to provide a symptomatic cure. The study by Meewhi Kim and Ilya Bezprozvanny proposed computational insights for selective targeting of M2 conformation of γ secretase complex with APP to reduce Ab42 load. This is an intriguing and original computational investigation that provides details on the conformational flexibility of the γ secretase complex with APP. However, there are a few minor corrections to be made before the final acceptance of the manuscript.
- “consistent with published biochemical analysis of APP processing at different subcellular locations”. This phrase occurs 4 times (line 30,70,258,278) in the article. In particular, either in the result section or in the discussion they should write in brief how consistent they (biochemical analysis in cited references) are in comparing with this study.
- In line 113, the reason why PS2 was not considered for such analysis is in a sentence or two.
- In figure 2 description, mention what is D1 and D2. Although in figure 3A it is clear (D1 and D2 structures) figure 2 proceeds figure 3. Either describe here or redirect to figure 3 in figure 2 description to ease readership.
- In figure 3d description …..shown relative to the catalytic aspartate D285 in line 211. Check this position it should be D385.
Minor corrections:
- Mention the name of web-based program in line 83( I see in methods but it is better to mention here as well).
- Fig 1A, depict the amino and carboxy terminal end
- Minor English correction- Check for the typo: “pay or play” in line 13, “Modelling or Modeling” (use the same word throughout), “Mut1 and Mut2 or MUT1 and MUT2” (use the same format throughout), in line 200 “structur or structure”
Author Response
We are thankful that referee found our study “intriguing and original” and provided a number of constructive comments that we addressed below.
- “consistent with published biochemical analysis of APP processing at different subcellular locations”. This phrase occurs 4 times (line 30,70,258,278) in the article. In particular, either in the result section or in the discussion they should write in brief how consistent they (biochemical analysis in cited references) are in comparing with this study.
Our apologies, we should have provide more information. We now included a paragraph that describes key biochemical results reported in the published literature regarding differences in Aβ40 and Aβ42 production in early and late endosomes. Specifically, we stated (lines 309-314) in Discussion:
"Our hypothesis is also consistent with biochemical analysis of APP processing in different subcellular locations. Indeed, it has been reported that late endosome com-partments contain Aβ peptides with higher 42/40 ratio [26-31]. Mechanistic studies of γ-secretase activity indicated increased Aβ42 recycling from early to late endosome [26]. However, these studies were focused on influence of endosomal pH, and our hypothesis regarding differences in membrane curvature requires further testing."
- In line 113, the reason why PS2 was not considered for such analysis is in a sentence or two.
PS2 was not included because CryoEM structures available to us are structures of PS1, this information is now added to the paper. It is possible to model PS2 structure as well, but in our opinion it will complicate presentation of this “conceptual” paper
- In figure 2 description, mention what is D1 and D2. Although in figure 3A it is clear (D1 and D2 structures) figure 2 proceeds figure 3. Either describe here or redirect to figure 3 in figure 2 description to ease readership.
Our apologies. We included this information and reference to Fig 3 as needed.
- In figure 3d description ….shown relative to the catalytic aspartate D285 in line 211. Check this position it should be D385.
Our apologies for the typo. Of course D385 is correct
Minor corrections:
- Mention the name of web-based program in line 83( I see in methods but it is better to mention here as well).
It is now stated that DISOPRED3 program described in ref 32 was used.
- Fig 1A, depict the amino and carboxy terminal end
Added to the figure 1.
- Minor English correction- Check for the typo: “pay or play” in line 13, “Modelling or Modeling” (use the same word throughout), “Mut1 and Mut2 or MUT1 and MUT2” (use the same format throughout), in line 200 “structur or structure”
Our apologies, corrected.
Reviewer 2 Report
The authors proposed the formation of complex between APP and gamma secretase in two conformation M1 and M2 according to structural analysis. I consider this work interesting and suitable for pubblication in this journal after some revisions.
1) In this paper, you modeled structures of PS1-gamma secretase using two CRYO-EM structures: 6iyc and 5fn3. I search these structures in Protein Data Bank and I noticed that the resolution of 5FN3 is very high, pheraphs too high for a speculative studies you carried out. Did you tried to use another structure for this purpose?
2) Which sofwtare did you use for modelling structures? Have you performed some steps of energy minimization after your modelling analysis? I think this was necessary to fully validate your models.
3) You suggested the possibile effects of membrane curvature for APP-Gamma secretase, but this is unclear to me how you built the membranes and how you place structures inside them. On the OPM (Orientation of Protein in Membranes) website, is stored the correct orientation of 6iyc in the membrane. Did you use such an orientation as your starting point? If not, I consider this section very speculative and it shoud be resized in the main text.
In Figure 1, line 103, please correct the sentence: "Domain structure of APP with locations of β- and γ-secretase cleavage sites indicated." to "Domain structure of APP with locations of β- and γ-secretase cleavage sites is indicated."
In Figure 3, please change "structur" to structure" (line 200). Remove one dot at the end of sentence in bold (line 199).
Line 228: please change "PDB" to "PDB_ids 6idf...."
Plase select another drawnig method (e.g. new cartoon) for Panels a,b and c of Figure 2. The information you describe is not fully observable in the Figure at this moment.
Author Response
We are very grateful that the reefree considers our work “interesting and suitable for publication in this journal”. Referee also had several comments related to our analysis that we address below.
1. In this paper, you modeled structures of PS1-gamma secretase using two CRYO-EM structures: 6iyc and 5fn3. I search these structures in Protein Data Bank and I noticed that the resolution of 5FN3 is very high, pheraphs too high for a speculative studies you carried out. Did you tried to use another structure for this purpose?
We indeed tested all available γ-secretase PDBs for the model building. In our hands 5FN3 provided most informative basis for the modeling as active site of γ-secretase was well resolved.
- Which software did you use for modelling structures? Have you performed some steps of energy minimization after your modelling analysis? I think this was necessary to fully validate your models.
As described in methods we build models using CC4 and Pymol. It should be stressed that our models are more “conceptual models” and we have not used specialized modeling software or performed energy minimization procedures. We generated these models by manually aligning the structures of γ-secretase in different conformation and APP α-helix. There is no doubt that these structural models can be further optimized and refined by energy minimization routines, but we would like to leave it to the experts in this type of modeling.
3. You suggested the possibile effects of membrane curvature for APP-Gamma secretase, but this is unclear to me how you built the membranes and how you place structures inside them. On the OPM (Orientation of Protein in Membranes) website, is stored the correct orientation of 6iyc in the membrane. Did you use such an orientation as your starting point? If not, I consider this section very speculative and it shoud be resized in the main text.
Thank you very much for this suggestion. We consulted OPM website and determined that structure of γ-secretase embedded in the membrane can indeed be modelled as a cylinder and orientation that we use is correct.
- In Figure 1, line 103, please correct the sentence: "Domain structure of APP with locations of β- and γ-secretase cleavage sites indicated." to "Domain structure of APP with locations of β- and γ-secretase cleavage sites is indicated."
Our apologies, corrected.
- In Figure 3, please change "structur" to structure" (line 200). Remove one dot at the end of sentence in bold (line 199).
Our apologies, corrected.
- Line 228: please change "PDB" to "PDB_ids 6idf...."
Our apologies, corrected.
- Plase select another drawnig method (e.g. new cartoon) for Panels a,b and c of Figure 2. The information you describe is not fully observable in the Figure at this moment.
Thank you for this suggestion. Figs 2b and 2c were redrawn as ribbon diagrams. In addition, and now added new panels 2d and 2e that display only TM6, TM7 and APP for models M1 and M2 for clarity.